# The Effect of Administration of a Phytobiotic Containing Cinnamon Oil and Citric Acid on the Metabolism, Immunity, and Growth Performance of Broiler Chickens

**DOI:** 10.3390/ani11020399

**Published:** 2021-02-04

**Authors:** Magdalena Krauze, Monika Cendrowska-Pinkosz, Paulius Matuseviĉius, Anna Stępniowska, Paweł Jurczak, Katarzyna Ognik

**Affiliations:** 1Department of Biochemistry and Toxicology, Faculty of Animal Sciences and Bio-Economy, University of Life Sciences in Lublin, Akademicka st. 13, 20-950 Lublin, Poland; anna.stepniowska@up.lublin.pl (A.S.); pawel15033@wp.pl (P.J.); katarzyna.ognik@up.lublin.pl (K.O.); 2Chair and Department of Human Anatomy, Medical University of Lublin, 20-090 Lublin, Poland; monika.cendrowska-pinkosz@umlub.pl; 3Department of Animal Nutrition, Lithuanian University of Health, Kaunas, Tilzes 18, LT-47181 Kaunas, Lithuania; Paulius.Matusevicius@lsmuni.lt

**Keywords:** phytobiotic, cinnamon oil, nutritional supplement, blood parameters, microbiological parameters

## Abstract

**Simple Summary:**

In poultry farming, additives are sought after to ensure the living needs of birds, improve the health of birds, and improve growth performance. Noticeably, more and more hopes for obtaining such effects are being placed in plant additives, called phytobiotics, which are safe as natural additives, both for the health of birds and for not leaving toxic residues in final products (meat and eggs). A valuable phytobiotic ingredient is cinnamon, used in the form of an oil or a powder, which is obtained from the bark or leaves of the cinnamon tree. Cinnamon oil can stimulate the appetite, increase the secretion of digestive enzymes, stimulate immunity, have anti-allergic and detoxifying properties, and also have a positive effect on reducing sugar levels in the body. It has antimicrobial properties that destroy the cell membranes of pathogens, and, thanks to its antioxidant properties, it accelerates wound healing and stimulates the functioning and regeneration of intestinal epithelial cells. It is also antiparasitic, especially against gastrointestinal parasites. Due to the number of valuable properties of cinnamon oil, it seems advisable to find the most favorable dosage and time of application of this component to the chickens’ water, that may cause intended effects such as improving the health of chickens and increasing the efficiency of their rearing.

**Abstract:**

It was postulated that a phytobiotic preparation containing cinnamon oil and citric acid added to drinking water for chickens in a suitable amount and for a suitable time would beneficially modify the microbiota composition and morphology of the small intestine, thereby improving immunity and growth performance without inducing metabolic disorders. The aim of the study was to establish the dosage and time of administration of such a phytobiotic that would have the most beneficial effect on the intestinal histology and microbiota, production results, and immune and metabolic status of broiler chickens. The experiment was carried out on 980 one-day-old male chickens until the age of 42 days. The chickens were assigned to seven experimental groups of 140 birds each (seven replications of 20 individuals each). The control group (G-C) did not receive the phytobiotic. Groups CT-0.05, CT-0.1, and CT-0.25 received the phytobiotic in their drinking water in the amount of 0.05, 0.1, and 0.2 mL/L, respectively, at days 1–42 of life (continuous application, CT). The birds in groups PT-0.05, PT-0.5, and PT-0.25 received the phytobiotic in the same amounts, but only at days 1–7, 15–21, and 29–35 of life (periodic application, PT). Selected antioxidant and biochemical parameters were determined in the blood of the chickens, as well as parameters of immune status and redox status. The morphology of the intestinal epithelium, composition of the microbiome, and production parameters of chickens receiving the phytobiotic in their drinking water were determined as well. The addition of a phytobiotic containing cinnamon oil and citric acid to the drinking water of broiler chickens at a suitable dosage and for a suitable time can beneficially modify the microbiome composition and morphometry of the small intestine (total number of fungi *p* < 0.001, total number of aerobic bacteria *p* < 0.001; and total number of coliform bacteria *p* < 0.001 was decreased) improving the immunity and growth performance of the chickens (there occurred a villi lengthening *p* = 0.002 and crypts deepening *p* = 0.003). Among the three tested dosages (0.05, 0.1, and 0.25 mL/L of water) of the preparation containing cinnamon oil, the dosage of 0.25 mL/L of water administered for 42 days proved to be most beneficial. Chickens receiving the phytobiotic in the amount of 0.25 mL/L had better growth performance, which was linked to the beneficial effect of the preparation on the microbiome of the small intestine, metabolism (the HDL level *p* = 0.017 was increased; and a decreased level of total cholesterol (TC) *p* = 0.018 and nonesterified fatty acids (NEFA) *p* = 0.007, LDL *p* = 0.041, as well as triacylglycerols (TAG) *p* = 0.014), and immune (the level of lysozyme *p* = 0.041 was increased, as well as the percentage of phagocytic cells *p* = 0.034, phagocytosis index *p* = 0.038, and Ig-A level *p* = 0.031) and antioxidant system (the level of LOOH *p* < 0.001, MDA *p* = 0.002, and the activity of Catalase (CAT) *p* < 0.001 were decreased, but the level of ferric reducing ability of plasma (FRAP) *p* = 0.029, glutathione *p* = 0.045 and vitamin C *p* = 0.021 were increased).

## 1. Introduction

In poultry farming, additives are sought after to improve the health and growth performance of birds while meeting their nutritional requirements. Phytobiotics play a growing role as potential alternatives to antibiotic growth promoters (AGP), because they are natural, easily available, nontoxic, and residue-free [1,2,3]. Phytobiotic materials can be herbal extracts or plant parts (leaves, rhizomes, roots, flowers, bark, or, less commonly, seeds) with the greatest accumulation of bioactive substances. Phytobiotic additives can stimulate the appetite, increase secretion of digestive enzymes, stimulate immunity, and exert bactericidal, antiviral, and antioxidant effects, as well as improve growth performance and the quality of animal products [4,5,6].

Cinnamon (*Cinnamomum cassia*) contains numerous essential oils, cinnamic acid, cinnamon aldehyde, caryophyllene oxide, eugenol, L-borneol, and many other valuable biologically active substances [7]. Cinnamon oil obtained from the bark or leaves of the cinnamon tree can be used as a preparation regulating digestion and increasing immunity, as well as exerting detoxifying, hypoglycaemic, and anti-inflammatory effects [4,8]. Furthermore, by protecting the building blocks and enzymatic components of cells against oxidation, it prevents changes in the structure and metabolic activity of these biomolecules. Cinnamon oil also has antimicrobial activity against bacteria (*Parahaemolyticus*, *Staphylococcus epidermis*, *Enterococcus faecalis*, *Pseudomonas aeruginosa*, *Salmonella* sp., *Staphylococcus aureus*, and *Escherichia coli*), pathogenic fungi, and molds [2]. The components of cinnamon oil destroy the cell membranes of pathogens, causing them to become damaged and disintegrate [9,10]. Owing to its antioxidant activity, cinnamon oil accelerates the healing of wounds and stimulates the functioning and regeneration of erythrocytes. It also has antiparasitic effects, especially against parasites of the digestive tract. Cinnamon oil can stimulate metabolism of carbohydrates and lipids and also exert anti-allergenic effects [4]. In addition to phytogenic additives, poultry rearing can also use organic acids, alone or combined with phytobiotics. According Fascina et al. [11], the addition of cinnamon oil and citric acid improves the nutrient digestibility of the diet and replaces the growth-promoting antibiotics. The results of the research by Pirgozliev et al. [12] and Ao et al. [13] showed that the combination of cinnamon oil and citric acid has a positive effect on the production performance of poultry as it lowers the pH of the intestinal contents and increases intolerance of bacterial growth to pH changes [12,13]. The result is better gut health, better intestinal villi integrity, and maximum nutrient absorption [14]. Additionally, un-dissociated organic acids can penetrate the lipid membrane of the bacterial cell and lower the pH inside it, which leads to the death of the bacterial cell [15]. Research by Ao et al. [13] and Rizzo et al. [16] showed that citric acid alone can also increase the ability to metabolize feed coefficients of metabolizable nutrients, which results in improved intestinal villi integrity and increased lipid absorption from the diet. According to Fascina et al. [11], phytobiotics significantly better improve the health of chickens in the initial phase of feeding, thanks to the higher production of pancreatic enzymes. Acidifying, on the other hand, improves nutrient metabolism in the growth phase much more, according to Fascina et al. [11], due to the presence of more intestinal contents. According the opinion of Fascina et al. [11], the combination of cinnamon oil and citric acid in the diet supports muscle growth, without much wear and tear of renewing tissues.

The phytobiotic preparation used in the experiment is a commercial preparation for poultry. The manufacturer assumes a dosage of the preparation in the amount of 0.1–0.2 mL/L of water, with the possibility of using it throughout the rearing period. For the purposes of the experiment, it was decided to check what health and production effect would be achieved by using the dose recommended by the manufacturer (0.1 mL/L), compared to a dose lower than 0.05 mL/L and greater than 0.25 mL/L than recommended by the manufacturer. Moreover, the introduction of two modes of administration, continuous and periodic, of the test product to the experimental scheme was aimed at checking whether the periodic application of phytobiotic supplement would bring comparable effects to the continuous application. The periodic application of the preparation was adapted to the feeding periods of the birds (start of the starter phase, 1–7 days; end of the starter phase, 15–21 days; and end of the grower phase, 29–35 days).

It was postulated that the addition of a phytobiotic preparation containing cinnamon oil and citric acid to drinking water for chickens in a suitable amount and for a suitable time would beneficially modify the microbial composition and morphometry of the small intestine, thereby improving their metabolism, immunity, and growth performance. The aim of the study was to determine what dosage of the phytobiotic and what duration of administration would have the most beneficial effect on the histology and microbiological composition of the intestine, immune status, and growth performance of broiler chickens.

## 2. Material and Methods

### 2.1. Phytobiotic

A widely available commercial feed additive was used in the study. The phytobiotic supplement contains 3000 mg/L of cinnamon oil and 150 mg/L of citric acid (EW Nutrition, Visbek, Germany).

### 2.2. Chemical Analysis of Volatile Oils in the Phytobiotic Preparation

Three samples of the phytobiotic were analyzed in a laboratory by capillary gas chromatography (GC). GC analysis was carried out using an Agilent 7890A GC System equipped with an FID and an Agilent 7683B Series auto-injector.

### 2.3. Animals

#### Experimental Plan

The experiment was carried out on one-day-old male Ross 308 broiler chickens raised until the age of 42 days. The birds were kept on straw litter in standard conditions in a building with regulated temperature and humidity. The size of the pens was in accordance with the Ross Broiler Management Manual, according to Aviagen [17]. The chickens had permanent access to drinking water and received ad libitum complete compound feeds, whose composition was established according to [18] (Table 1).

The experimental procedure was approved by the Second Local Ethics Committee for Experiments with Animals in Lublin (approval no. 38/2018). The experimental design for administration of the phytobiotic preparation is shown in Table 2. The experiment was carried out on 980 chickens assigned to seven experimental groups of 140 birds each (seven replications of 20 individuals each). The control group (G-C) did not receive the phytobiotic. The phytobiotic preparation containing cinnamon oil was administered in two different application modes: continuous (CT—continuous application) and periodic (PT—periodic application). Groups CT-0.05, CT-0.1, and CT-0.25 received the phytobiotic in their drinking water in the amount of 0.05, 0.1, and 0.25 mL/L, respectively, at days 1–42 of life. The birds in groups PT-0.05, PT-0.5 and PT-0.25 received the probiotic in the same amounts, but only at days 1–7, 15–21, and 29–35 of life (21 days in total).

Body weight (BW) and feed intake were monitored at the end of each week and used to calculate the feed conversion ratio (FCR). Mortality was also monitored during the experiment. After six weeks of rearing, 21 birds from each group were slaughtered and dissected. At 42 days of age, 21 broilers per group (three birds representing the average body weight in each pen) were slaughtered at a slaughterhouse. The birds (without being transported) were electrically stunned (400 mA, 350 Hz), hung on a shackle line, and exsanguinated by a unilateral neck cut severing the right carotid artery and jugular vein. After a 3-min bleeding period, the birds were scalded at 61 °C for 60 s, de-feathered in a rotary drum picker for 25 s, and manually eviscerated.

### 2.4. Laboratory Analysis

At 42 days of age, 21 birds (3 each from each replicate group) representing the average body weight of each group were selected and fasted for 8 h. Blood samples for further analysis were taken from the wing vein of 14 birds from each group into two separate tubes. The first tube, for the acquisition of plasma, contained heparin as an anticoagulant, while the second tube, for serum, did not contain an anticoagulant. The blood samples were centrifuged at 3000× *g* for 10 min. Then, they were cooled and analyzed within 4 h of collection. Kits developed by Cormay (Poland) were used to determine biochemical parameters in the plasma: glucose (GLU), total protein (TP), urea (UREA), total cholesterol (TC), uric acid (UA), bilirubin (BIL), creatinine (CREAT), high-density (HDL) and low-density (LDL) cholesterol, and triacylglycerols (TAG). The activity of the following enzymes was determined in the plasma: alanine aminotransferase (ALT; EC 2.6.1.2), aspartate aminotransferase (AST; EC 2.6.1.1), creatinine kinase (CK; EC 2.3.7.2), lactate dehydrogenase (LDH; EC 1.1.1.27), γ-glutamyltransferase (GGT; EC 2.3.2.2), alkaline phosphatase (ALP; EC 3.1.3.1), acidic phosphatase (AC; EC 3.1.3.2) and 3-hydroxybutyrate dehydrogenase (HBDH; EC 1.1.1.30). The level of nonesterified fatty acids (NEFA) was determined using reagents by Randox (Germany). The activity of antioxidant enzymes in the plasma was analyzed using spectrophotometric assays. The ferric reducing ability of plasma (FRAP), which represents total antioxidant capacity, was determined according to Benzie and Strain [19]. The level of lipid hydroperoxides (LOOH) was determined according to Gay and Gębicki [20]. Catalase (CAT, EC 1.11.1.6), (SOD, EC 1.15.1.1), activity and levels of immunoglobulin A (IgA), interleukin 6 (IL-6), malondialdehyde (MDA), ascorbic acid (Vit. C), and glutathione (GSH + GSSG) were determined by an immune-enzymatic ELISA assay using kits from Elabscience Biotechnology Co., Ltd. (Houston, TX, USA). To determine the phagocytic activity of leukocytes against *Staphylococcus aureus* strain 209P, the percentage of phagocytic cells, the phagocytic index, and serum lysozyme activity were assessed by the turbidimetric method according to Siwicki and Anderson [21]. The ability to kill phagocytized bacteria was assessed based on the capacity of neutrophils to produce oxygen radicals, using the nitroblue tetrazolium test [22].

After dissection, samples of the jejunum were collected and placed in sterile containers. The material was then subjected to microbiological analysis to determine the number of aerobic bacteria on nutrient agar, the number of coliform bacteria on violet-red bile lactose agar (VRBL) (incubation for 24 h at 37 °C), and the total number of yeasts and molds on DG18 medium (incubation for 5–7 days at 25 °C). After incubation, the total bacterial cell count was determined, and the values obtained were expressed as CFU/g. Microbiological testing included macroscopic and microscopic evaluations as well as Gram staining. The final identification of the bacterial colonies was performed using API tests (bioMérieux, Warszawa, Poland), according to PN-ISO 4832, PN-EN ISO 7218, and PN ISO 4832. Intestinal samples collected during dissection were subjected to histological evaluation. From each intestinal sample, 20 intestinal villi were selected. A representative section 2 cm in length, cut 1 cm behind Meckel’s diverticulum toward the caecum, was collected for histological examination. Its length was measured from the tip to the base, and then it was cut into slices. The depth of 20 crypts was measured as well, after which they were cut in two lengthwise. All intestinal segments were fixed in a 4% buffered formalin (Sigma-Aldrich Corporation, St. Louis, MO, USA) solution with pH 7.2 for 24 h and then stored in 70% ethanol. After the fixation of intestinal fragments, they were placed in increasing concentrations of alcohol solutions (70%, 95%, absolute) for dehydration. Then the samples were cleaned with xylene and embedded in paraffin blocks in a tissue processor (Leica TP 1020). Each intestinal histomorphological tissue sample was prepared and stained with haematoxylin and eosin solution using standard paraffin embedding methods [23]. A computerized microscopic-image-analysis system was used to estimate villus length and crypt depth. A light microscope (Nikon Eclipse E600, Tokyo, Japan) with a digital camera (Nikon DS-Fi1, Tokyo, Japan) and a PC with image analysis software (NIS-Elements BR-2.20, Tokyo, Japan, laboratory imaging) were used.

### 2.5. Statistical Analysis

The model assumptions of normality and homogeneity of variance were verified by the Shapiro–Wilk and Levene tests, respectively. The results were analyzed by one-way ANOVA. Comparison of the control (G-C) vs. all other groups was performed by planned contrast analysis. In addition, Dunnett’s two-tailed post hoc test was used to compare each experimental group with group G-C. In a model without group G-C, two-way ANOVA was performed to examine the following effects: D—dose effect, T—time effect, and DxT—interaction between dose and time. When a significant interaction effect was noted, the Newman–Keuls test was used to determine the differences between the factors. The statistical analysis was performed according to the GLM procedure in Statistica 13 PL software (StatSoft Corp^®^, Tulsa, OK, USA). Treatment effects were considered to be significant at *p* ≤ 0.05. All data were expressed as mean values with pooled SE.

## 3. Results

The results of the analysis revealed that the phytobiotic preparation with cinnamon oil and citric acid contained 78.08% cinnamaldehyde, 2.17% cinnamyl acetate, 0.09% camphene, and 3.25% eugenol. Detailed information on the intake of the phytobiotic preparation by the chickens with their drinking water is presented in Table 2.

Contrast analysis showed that the levels of LOOH and MDA in the chickens from all groups receiving the phytobiotic were lower (*p* < 0.001; *p* = 0.002) than in group G-C (Table 3).

Catalase activity in the chickens from treatments CT-0.05, CT-0.1, and CT-0.25 was higher than in group G-C (*p* < 0.001). ALT activity in the plasma of chickens from treatments CT-0.05, CT-0.1, and PT-0.25 was lower than in the control (*p* = 0.025). In the case of treatment CT-0.25, ALT activity was higher (*p* = 0.025) than in G-C. In the blood of chickens from all groups receiving the phytobiotic (irrespective of the dosage), the FRAP value was significantly higher (*p* = 0.029) than in group G-C (Table 4). In the plasma of chickens from the CT-0.1 and CT-0.25 treatments, as well as from all treatments in which the phytobiotic was administered periodically (PT), higher concentrations of GSH + GSSH (*p* = 0.045) and vitamin C (*p* = 0.021) were noted than in group G-C. The UA level in the blood of the chickens from the CT-0.25 treatment was lower (*p* = 0.027) than in G-C.

One-way analysis showed that the addition of the phytobiotic decreased the TC level in the blood of the chickens from both the CT and PT treatments relative to the control (Table 5). In the blood of chickens from the CT, PT-0.05, and PT-0.25 treatments, an increase (*p* = 0.017) was noted in HDL cholesterol relative to the control. In the blood of chickens from the CT, PT-0.05, and PT-0.25 treatments, a decrease was observed in the concentration of TAG (*p* = 0.014) and in the level of NEFA (*p* = 0.007) relative to group G-C.

In the blood of the chickens from all groups receiving the phytobiotic, there was a decrease (*p* < 0.001) in LDH activity relative to group G-C (Table 6).

GGT activity in the blood of chickens from the CT-0.05, CT-0.25, and PT-0.25 treatments was lower (*p* = 0.034) than in the control. The use of the phytobiotic reduced (*p* = 0.021) plasma CK activity in all treatments relative to G-C. Comparison with the effects obtained in the control group also revealed that the addition of the phytobiotic increased (*p* = 0.028) HBDH activity in the blood of chickens from all CT treatments, while in the PT treatments this effect was obtained following administration of 0.1 and 0.25 mL/l. In the case of AC, activity was higher (*p* = 0.025) in the chickens from the CT-0.05 treatment than in the control but lower in groups PT-0.05 and PT-0.1. The lysozyme level was increased (*p* = 0.041) in the blood of chickens from all treatments with the phytobiotic in comparison with the control group Table 7.

In the case of the percentage of phagocytic cells, all dosages of the phytobiotic, except 0.05 mL/L, caused an increase (*p* = 0.034) in the percentage of these cells relative to the control. In addition, only the dosage of 0.25 mL/L, applied continuously or periodically, caused an increase (*p* = 0.038) in the phagocytic index in the blood of the chickens relative to the control. In the case of IL-6, only the dosage of 0.25 mL/L did not increase its level in the blood, while the other dosages caused a marked increase (*p* = 0.014) in its concentration compared to the control. A pronounced increase in the IgA level compared to the control was noted in the blood of chickens receiving continuous or periodic supplementation of the phytobiotic at 0.1 or 0.25 mL/L (*p* = 0.031).

In the chickens from the CT-0.1, CT-0.25, and PT-0.25 treatments, final body weight at 42 days of age was higher (*p* = 0.025) than in group G-C (Table 8). In comparison to G-C, lower FCR values were noted in chickens from the CT-0.1, CT-0.25, and PT-0.25 treatments (*p* = 0.031). Mortality of chickens in the experiment was 1% in groups CT-0.1, CT-0.25, and PT-0.25, and 3% in G-C.

### 3.1. Effect of Dose

In the case of the total number of aerobic bacteria, a dose–time interaction was observed, as the count was highest (*p* = 0.008) in the intestinal contents of chickens receiving the phytobiotic in the amount of 0.1 mL/L, irrespective of the duration of administration, which was not observed for doses of 0.05 and 0.25 mL/L. In addition, irrespective of the duration of application, the dose of 0.25 mL/L caused a marked increase in IgA (*p* = 0.031) in the blood of the broiler chickens (Table 9).

The use of different doses of a phytobiotic containing citric acid caused variation in the biochemical parameters of the blood of the chickens and in the number of aerobic bacteria in the contents of the small intestine. As the dosage of the phytobiotic increased, a decrease (*p* < 0.001) was observed in the level of LOOH in the blood (Table 3). A dose–time interaction (*p* < 0.001) was noted for CAT activity in the blood, as it was lower in chickens receiving the phytobiotic in the amount of 0.05 or 0.1 mL/L, irrespective of the duration of application, which was not observed for the dose of 0.25 mL/L. Analysis of the level of lipid status parameters showed that irrespective of the duration of application, the blood of chickens receiving the phytobiotic at a dose of 0.25 mL/L had a lower (*p* = 0.034) level of NEFA than in birds receiving the additive in the amount of 0.05 or 0.1 mL/L (Table 5). In the case of CK activity, there was a dose–time interaction (*p* = 0.042), as the effect of the intermediate dosage of the phytobiotic (0.1 mL/L) was seen as a decrease in the activity of this enzyme, while the other doses (0.05 and 0.25 mL/L) did not have this effect. The doses of 0.05 and 0.25 mL/L increased activity of CK (*p* = 0.042).

### 3.2. Effect of Time

A significant dose–time interaction was shown for the number of aerobic bacteria in the contents of the small intestine (*p* = 0.008). The interaction resulted from the marked increase in this parameter in the CT treatments but only for the dose of 0.1 mL/L. The number of aerobic bacteria in the groups receiving the phytobiotic periodically (PT) at doses of 0.05 and 0.25 mL/L did not differ statistically.

The use of two different durations of application of the phytobiotic (continuous and periodic) led to differences in the antioxidant status, redox status, lipid profile, and activity of selected enzymes in the blood of the broiler chickens. Irrespective of the dose used, the longer application time (CT) decreased the concentrations of LOOH and MDA (*p* = 0.004; *p* = 0.001) in the plasma (Table 3). Continuous administration of the phytobiotic (CT) caused a greater increase in CAT activity (*p* < 0.001) than periodic use (PT).

Analysis of enzyme activity in the blood of the chickens showed that irrespective of the dosage of the phytobiotic, longer administration (CT) caused a decrease in the activity of LDH and HBDH (*p* = 0.030; *p* = 0.021) compared to the result obtained following periodic application of the phytobiotic. A dose–time interaction (*p* = 0.042) was noted for CK activity, as the effect of continuous application of the phytobiotic was manifested as a reduction in the activity of this enzyme, while this effect was not observed in the case of PT. Continuous application of the phytobiotic (CT) reduced activity of HBDH (*p* = 0.049) more than PT application (Table 6).

## 4. Discussion

Numerous scientific studies focus on the potential use of various plant supplements in the diet of chickens to improve their growth performance without adversely affecting their health [24]. One condition of good health in birds is a state of eubiosis in the intestines, which in turn is conditioned by a balance between useful and undesirable bacteria in the microbiome [25]. In the present study, the use of cinnamon oil had a beneficial effect in the form of increased villus length and crypt depth in the small intestine of broiler chickens. The most beneficial results were obtained in the groups receiving the highest doses of the phytobiotic (0.1 and 0.25 mL/L), especially when it was administered continuously, i.e., from 1 to 42 days of rearing. An improvement in the intestinal histomorphology of chickens following the use of cinnamon preparations has also been reported by Wasman [26] (0.5 mL cinnamon oil/kg feed), by Chowdhury et al. [27] (300 mg cinnamon bark oil on kg feed), by Yang et al. [28] (100 mg cinnamon aldehyde per kg feed), and by Mahmoud et al. [29] (dichloromethane extract of cinnamon in dose of 10 and 20 mg/kg/day). The inclusion of cinnamon oil in the diet in amounts 250 or 500 mg/kg feed improve the activity of antioxidant mechanisms in Japanese quails [30]. According to Biasato et al. [31], the integrity of the small intestine is determined by appropriate villus height and crypt depth, with longer villi increasing the absorptive surface area, which is conducive to absorption of nutrients from the intestinal contents [32]. According to Reis et al. [33], the components of cinnamon oil improve enterocyte viability and reduce oxidative damage to the intestinal epithelium, leading to better nutrient absorption and improving the birds’ overall condition. In our study, the addition of a phytobiotic containing cinnamon oil increased the number of aerobic bacteria in the intestinal contents of the broiler chickens and significantly reduced the number of fungi and coliforms. Results obtained by Mehdipour and Afsharmanesh [34] indicate that cinnamon oil (200 ppm/kg diet) can limit the growth and colonization of numerous pathogenic and nonpathogenic species of bacteria in the intestines by damaging their cell membrane, causing the cells of pathogens to disintegrate. According to Reis et al. [33], the main active substance in cinnamon oil, cinnamon aldehyde, can selectively inhibit the growth and development of both pathogenic and commensal intestinal bacteria, which may help to balance the microbial population and improve intestinal health. Yang et al. [28], after administering cinnamon oil (100 mg/kg feed) to chickens, also observed an improvement in the intestinal microbiome, with a marked increase in the number of beneficial bacteria. A decrease in the number of coliforms in the intestinal contents following the use of cinnamon oil (25–100 µg/mL) has been reported by Gupta et al. [34]. In the research, Abramowicz et al. [35] compared the effectiveness of the preparation containing cinnamon oil (0.25 mL/L) in the context of the effect of the supplement containing *Bacillus subtilis* on intestinal morphometry of broiler chickens. The results of using both additives were similar, both preparations elongated the villi and deepened the crypts. Cinnamon oil stimulates mucus secretion in the intestines, which additionally limits adhesion of pathogenic bacteria to the enterocytes. Furthermore, the components of cinnamon oil disturb the balance of inorganic ions in the living environment of microbes, which promotes disintegration of the bacterial cell membrane [2].

An important indication of the beneficial effect of cinnamon oil on broiler chickens may be the marked improvement in parameters of antioxidant defence and redox status observed in the present study. The addition of cinnamon oil caused a beneficial increase in the levels of FRAP, Vit. C, and GSH + GSSH, accompanied by a decrease in the concentrations of unfavourable oxidation parameters, i.e., MDA and LOOH. A beneficial decrease in the MDA level and an increase in the FRAP value in the blood of chickens receiving cinnamon oil has also been observed by Yang et al. [28], Faix et al. [29], and Symeon et al. [36]. Keshvari et al. [37] also believe that the administration of cinnamon oil to chickens can inhibit the formation of MDA and weaken lipid-peroxidation reactions and the unfavorable effects of free radicals. According to Abd El-Hack et al. [2], many biologically active substances, including eugenol, linalool, and cinnamon aldehyde, can significantly slow down lipid-peroxidation reactions. This prevents unfavorable oxidation of cell membrane components and free radical generation, thereby limiting the formation of MDA [2]. Symeon et al. [36] state that the antioxidant effect of substances in cinnamon (0.5 or 1.0 mL cinnamon oil per kg diet), apart from inhibition of lipid peroxidation, may also result from an increase in the activity of GPx and GGT, responsible for antioxidant defence [38]. According to the authors, diet supplementation with phytobiotics can be a good means of regular introduction of a natural antioxidant to phospholipid membranes and inhibition of oxidation reactions. According to Lee et al. [39], the components of cinnamon oil reduce the intensity of lipid peroxidation, especially of polyunsaturated fatty acids, by enhancing the activity of antioxidant liver enzymes. Faix et al. [40] also state that the components of cinnamon oil increase the activity of antioxidant enzymes in the liver, thereby inhibiting lipid peroxidation. The addition of cinnamon oil also improves liver metabolism, resulting in an increase in the level of polyunsaturated fatty acids and omega-6 fatty acids, accompanied by a decrease in saturated fatty acids, in both the liver tissue and the blood [36]. Fki et al. [41], in a study of rats, demonstrated that the phenolic compounds in essential cinnamon oils increase CAT activity, which neutralizes hydrogen peroxide and converts lipid hydroperoxides into nontoxic substances. In the present study, continuous administration of the phytobiotic, irrespective of the dose, increased CAT activity while having no effect on SOD activity, which is consistent with results reported by Fki et al. [41], Faix et al. [36], Ciftci et al. [42], and Al-Kassie [43]. The increase in CAT activity should be considered a beneficial effect, as it may suggest a decrease in the severity of stress reactions in the cells [36]. According to Sadeghi et al. [44], who observed that the addition of cinnamon powder to the diet of chickens increases CAT and ALP activity as well as the FRAP value, this additive significantly improves the antioxidant activity of blood serum.

The present study showed that the addition of cinnamon oil also has a beneficial effect on the metabolic profile of the blood of chickens. The addition of cinnamon oil caused a reduction in the cholesterol concentration, increasing HDL cholesterol and decreasing the LDL fraction in the blood of the chickens. This effect was observed irrespective of the dose of the phytobiotic and the duration of its administration. A similar result was obtained by Toghyani et al. [45]. In the present study, cinnamon oil also caused a decrease in the TAG concentration and activity of NEFA in the blood of chickens from all treatments, except the one in which the smallest dose of the phytobiotic (0.05 mL/L) was administered periodically. According to Faix et al. [36] and Lee et al. [39], cinnamon oil, especially cinnamic acid and its derivatives, effectively inhibits the activity of 3-hydroxy-3-methyl-glutaryl-CoA (HMG-CoA) reductase, thereby decreasing the cholesterol level in the blood. Kriaa et al. [46] state that the bacteria in the intestinal microbiome utilize cholesterol from the intestinal contents to build their own cell walls, thereby reducing the amount of exogenous cholesterol in the body. Koochaksaraie et al. [8] believe that cinnamon aldehyde is primarily responsible for the hypocholesterolaemic effect of cinnamon oil, by stimulating excretion of cholesterol from the body. According to Karimi-Kivi et al. [47], a decrease in the TAG and NEFA level in the blood confirms that cinnamon oil beneficially reduces lipolysis, which is physiologically accompanied by an increase in the NEFA level. The components of the oil cause intensive utilization of glycerol for re-synthesis of glucose in the gluconeogenesis process, thus limiting synthesis of TAG [47]. On the other hand, an increase in the NEFA level reduces the ability of cells to store TAG or to use them as an energy source, leading to lipotoxicity and the activation of inflammatory processes and oxidative stress [47]. Sarica et al. [48] suggest that a diet for quails enriched with cinnamon oil as well as prebiotics or probiotics can beneficially decrease the level of total cholesterol and triacylglycerols in the blood. A beneficial effect in studies on poultry has also been obtained by Dev et al. [49] and by Ognik and Krauze [50] using prebiotics, as well as by Krauze et al. [51] using cinnamon oil and probiotics in the diet of broiler chickens. In the present study, the decrease in ALT activity induced by periodic administration of all doses of the phytobiotic and by continuous application of the highest dose (0.25 mL/L) can be regarded as a beneficial effect. Faix et al. [36], Ciftci et al. [42], and Al-Kassie [43], whose research has also shown a decrease in ALT activity, express similar opinions. The authors state that the decrease in the activity of this aminotransferase may suggest that cinnamon oil has a significant hepatoprotective effect [36]. In our study, the addition of cinnamon oil to the diet of chickens also markedly decreased the activity of GGT, CK, LDH, and ALP. Similar results of the use of cinnamon oil in chicken diets have been reported by Tabatabaei et al. [52], notably in chickens infected with *E. coli* as well. Administration of the phytobiotic containing cinnamon oil also improved immune parameters. All doses of the phytobiotic, administered continuously or periodically, caused a beneficial decrease in the level of IL-6 and an increase in the lysozyme concentration in the blood of the broiler chickens. Administration of the phytobiotic in the amount of 0.1 or 0.25 mL/L reduced the level of IgA in the blood of the chickens. A similar effect was obtained with cinnamon powder by Sang-Oh et al. [5], who demonstrated that the addition of cinnamon increases the level of IgA in the blood of chickens and increases the weight of immune organs. According to the authors, the improvement in immune status in combination with the antibacterial and antioxidant effects of the components of cinnamon help to reduce mortality and the incidence of diarrhoea in chicks, as well as to increase the final body weight of broiler chickens. In our opinion, the addition of a phytobiotic containing cinnamon oil also improved indicators of the phagocytic activity of heterophils. Similar results in studies using cinnamon oil in the diet of chickens have been reported by Yang et al. [28], Faix et al. [36], Ciftci et al. [42], and Al-Kassie [43]. According to Faix et al. [36], cinnamon oil exerts immunomodulatory effects by stimulating nonspecific immunity, i.e., the functions of macrophages. Acting as a phagocytic, bactericidal, and antitumour effector, cinnamon oil stimulates interactions with lymphocytes and macrophages, thereby initiating and regulating the immune response [28,35].

In the present study, the greatest improvement in growth performance, correlated with the most favorable feed conversion ratio (FCR), was observed after both continuous (42 days) and periodic administration of the highest dose of the phytobiotic (0.25 mL/L) in the drinking water, and after continuous administration at a dose of 0.1 mL/L. Although the most beneficial rearing effects were obtained following continuous administration at 0.25 mL/L, it should be noted that continuous administration of the preparation at a dose of 0.1 mL/L resulted in comparable and equally satisfactory growth performance as periodic administration at a dose of 0.25 mL/L, while reducing consumption of the phytobiotic by nearly 11%.

Many researchers have tested various cinnamon-based preparations in the diet of poultry and obtained satisfactory growth performance. Dietary supplementation with cinnamon oil has been shown to improve the growth performance of broiler chickens [41,44,50,52] and quails [32,47], as well as resulting in a more favorable feed conversion ratio [32,53,54,55]. According to Milind and Deepa [56], cinnamon oil contains numerous volatile compounds, as well as saturated and unsaturated fatty acids, which improve rearing efficiency and FCR and reduce mortality in chickens. Abd El-Hack et al. [2] believe that cinnamon can be used as a potential alternative to antibiotics in production conditions in the poultry industry. In addition, according to Gomathi et al. [57], enrichment of the diet with cinnamon-based additives beneficially increases the amount of unsaturated fatty acids in the blood and muscle, thus reducing the pool of saturated acids. Lee et al. [39], however, point out that diet supplementation with cinnamon aldehyde can reduce water intake by chickens, especially females, which can be dangerous in the case of laying hens. According to Wasman et al. [27], the improvement in rearing efficiency and the quality of the final poultry products (meat and eggs) is due to the antibacterial and antifungal properties of cinnamon. The scientific literature, however, contains numerous reports indicating that phytobiotics do not significantly affect growth performance, the survival rate of poultry, or FCR [28,36,58,59,60,61].

## 5. Conclusions

Of the three doses (0.05, 0.1 and 0.25 mL/L water) of the phytobiotic containing cinnamon oil administered to chickens, the most beneficial was 0.25 mL/L water administered for 42 days. Chickens receiving the phytobiotic at 0.25 mL/L had the best growth performance, which was linked to the beneficial effect of the preparation on the microbiome and morphometry of the small intestine, on the metabolism, and on the immune and antioxidant systems.

## Figures and Tables

**Table 1 animals-11-00399-t001:** Composition of diets for chickens (g/kg).

Ingredients	Starter 1–3 Week	Grower 4–5 Week	Finisher 6 Week
Wheat	452.8	367.63	330.70
Maize	150.0	250.0	300.0
Soybean meal 46% protein	272.21	227.90	178.09
Rapeseed meal 37% protein	20.0	40.0	60.0
Soybean oil	20.0	40.0	60.0
DDGS ^1^ 26% protein	40.07	43.58	46.87
Monocalcium phosphate	11.03	5.42	2.05
Coarse-grained fodder chalk ^2^	-	10.93	8.52
Fine-grained fodder chalk	16.07	-	-
NaCl	3.63	3.23	2.83
DL-methionine 99%	3.61	2.40	2.00
L-Lysine HCl	4.27	2.97	3.12
L-threonine 99%	1.31	0.94	0.82
Premix ^3,4^	5	5	5
	Calculated nutrient composition of diet (g/kg) ^5^		
Crude protein	210.0	198.5	187.5
Crude fibre	27.2	29.8	32.2
Crude fat	65.9	74.5	81.4
Lysine	13.5	11.7	10.9
Methionine	6.7	5.5	5.0
Methionine + Cysteine	10.1	8.8	8.3
Tryptophan	2.5	2.3	2.1
Arginine	13.1	12.1	11.1
Total calcium	9.8	7.3	6.0
Available phosphorus	3.9	2.8	2.1
Sodium	1.6	1.5	1.4
Metabolizable energy, kcal/kg	3070	3140	3190

^1^ DDGS-maize distillers dried grains with solubles. ^2^ Calcium carbonate. ^3^ Vitamin provided per kg of diet: weeks 1–3: vitamin A, 15,000 IU; vitamin D_3_, 5000 IU; vitamin E, 112 IU; vitamin K_3_, 4 mg; vitamin B_1_, 3 mg; vitamin B_2_, 8 mg; vitamin B_6_, 5 mg; vitamin B_12_, 16 mg; folic acid, 2 mg; biotin, 0.2 mg; nicotinic amid, 60 mg; calcium pantothenicum, 18 mg; choline, 1.8 g; weeks 4–5: vitamin A, 12,000 IU; vitamin D_3_, 5000 IU; vitamin E, 75 IU; vitamin K_3_, 2 mg; vitamin B_1_, 2 mg; vitamin B_2_, 6 mg; vitamin B_6_, 4 mg; vitamin B_12_, 16 mg; folic acid, 1.75 mg; biotin, 0.05 mg; nicotinic amid, 60 mg; calcium pantothenicum, 18 mg; choline, 1.6 g; week 6: vitamin A, 12,000 IU; vitamin D_3_, 5000 IU; vitamin E, 75 IU; vitamin K_3_, 2 mg; vitamin B_1_, 2 mg; vitamin B_2_, 5 mg; vitamin B_6_, 3 mg; vitamin B_12_, 11 mg; folic acid, 1.5 mg; biotin, 0.05 mg; nicotinic amid, 35 mg; calcium pantothenicum, 18 mg; and choline, 1.6 g. ^4^ Trace minerals provided per kg of diet: Mn, 100 mg; Zn, 80 mg; Fe, 80 mg; Cu, 8 mg; I, 1 mg; Se, 0.15 mg; coccidiostat-salinomycin (except week 6). ^5^ Calculated according to the Polish feedstuff analysis tables [18].

**Table 2 animals-11-00399-t002:** Experimental design of administration of the phytobiotic preparation to chickens.

	Treatment						
	G-C ^2^	CT-0.05 ^2^	CT-0.1 ^2^	CT-0.25 ^2^	PT-0.05 ^2^	PT-0.1 ^2^	PT-0.25 ^2^
Cycles of administration of phytobiotic contain cinnamon oil and citric acid ^1^	0	6 × 7	6 × 7	6 × 7	3 × 7	3 × 7	3 × 7
Total intake of phytobiotic, mL/bird	0	0.33	0.633	1.658	0.14	0.28	0.72
Total intake of cinnamon oil, mg/bird	0	0.99	1.899	4.974	0.42	0.84	2.16
Total intake of citric acid, mg/bird	0	0.495	0.949	2.487	0.21	0.42	0.108

^1^ 6 × 7—intake at days 1–42 of age; 3 × 7—intake at days 1–7, 15–21 and 29–35 of age ^2^ Treatments: G-C—group receiving water without a phytobiotic supplement; CT-0.05 and PT-0.05—groups receiving water with 0.05 g/L of the phytobiotic; CT-0.1 and PT-0.1—groups receiving water with 0.1 g/L of the phytobiotic; CT-.25 and PT-0.25—groups receiving water with 0.25 g/L of the phytobiotic. CT-0.05, CT-0.1, and CT-0.25—continuous administration of the phytobiotic from 1 to 42 days of rearing; PT-0.05, PT-0.1, and PT-0.25—periodic administration of the phytobiotic at days 1–7, 15–21, and 29–35 of rearing.

**Table 3 animals-11-00399-t003:** Effect of the level and duration of application of the phytobiotic containing cinnamon oil on redox status in the blood of the chickens ^1^.

		LOOH µmol/L	MDA µmol/L	SOD U/mL	CAT U/mL	AST U/L	ALT U/L
G-C ^2^		4.98	0.62	32.12	1.84	222.4	4.91
CT-0.05		3.71 *	0.44 *	26.14	2.59 *	224.8	4.41
CT-0.1		2.53 *	0.35 *	31.12	2.61 *	218.4	3.78 *
CT-0.25		2.12 *	0.33 *	28.91	2.65 *	226.7	2.55 *
PT-0.05		3.89 *	0.65 *	31.36	1.44 *	225.8	5.66 *
PT-0.1		3.94 *	0.54 *	29.69	1.47 *	222.3	4.42
PT-0.25		2.59 *	0.53 *	29.74	1.25 *	227.1	3.89 *
SEM		0.112	0.025	0.43	0.12	1.25	0.14
Dosage effect (D)	0.05 (ml/L)	3.8	0.545	28.75	2.02 ^a^	225.3	5.04
	0.1 (ml/L)	3.24	0.445	30.41	2.04 ^a^	220.4	4.1
	0.25 (ml/L)	2.36	0.43	29.33	1.95 ^b^	226.9	3.22
Time effect (T)	CT	2.787 ^b^	0.373 ^b^	28.72	2.62 ^a^	223.3	3.58
	PT	3.473 ^a^	0.573 ^a^	30.26	1.39 ^b^	225.07	4.66
*p*-value							
G-C vs. all other		<0.001	0.002	0.257	<0.001	<0.001	0.025
D effect		0.031	0.043	0.298	0.017	0.0131	0.037
T effect		0.004	0.001	0.398	<0.001	0.002	0.003
DxT interaction		0.054	0.021	0.131	<0.001	0.034	0.369

* Means within the same column differ significantly from the control at *p* ≤ 0.05. ^a,b^ Means within the same column differ significantly (*p* ≤ 0.05) according to Newman–Keuls mean comparison (only in the case of significant dose–time (DxT) interaction). ^1^ Data represent mean values of seven replications per treatment. SEM = SD divided by the square root of the replication number, *n* = 7. ^2^ Designations for the treatments as for Table 2.

**Table 4 animals-11-00399-t004:** Effect of the level and duration of application of the phytobiotic (containing cinnamon oil) on antioxidant status in the blood of the chickens ^1^.

		FRAP µmol/L	GSH + GSSH µmol/L	VIT. C mg/L	UA µmol/L	BIL µmol/L	CREAT µmol/L
G-C ^2^		74.2	0.045	0.32	148.49	5.78	21.47
CT-0.05		93.14 *	0.059	0.57	146.78	5.47	20.47
CT-0.1		99.84 *	0.066 *	0.71 *	144.22	5.47	20.98
CT-0.25		147.1 *	0.074 *	0.77 *	139.24 *	5.23	21.04
PT-0.05		89.14	0.062 *	0.56 *	147.25	5.47	22.01
PT-0.1		94.84 *	0.069	0.57 *	142.22	5.55	21.47
PT-0.25		97.72 *	0.069 *	0.71 *	144.27	5.74	21.47
SEM		5.22	0.002	0.014	7.74	0.28	0.47
Dosage effect (D)	0.05 (ml/L)	91.14	0.061	0.57	147.02	5.47	21.24
	0.1 (ml/L)	97.34	0.068	0.64	143.22	5.43	2123
	0.25 (ml/L)	122.41	0.072	0.74	141.55	5.49	21.26
Time effect (T)	CT	113.36	0.066	0.68	143.4	5.37	20.80
	PT	93.9	0.066	0.61	144.58	5.59	21.65
*p*-value							
G-C vs. all other		0.029	0.045	0.021	0.027	0.055	0.077
D effect		0.048	0.069	<0.001	0.039	0.044	0.578
T effect		0.068	0.081	<0.001	0.042	0.031	0.854
DxT interaction		0.069	0.374	0.458	0.287	0.174	0.125

* Means within the same column differ significantly from the control at *p* ≤ 0.05. ^1^ Data represent mean values of seven replications per treatment. SEM = SD divided by the square root of the replication number, *n* = 7. ^2^ Designations for the treatments as for Table 2.

**Table 5 animals-11-00399-t005:** Effect of the level and duration of application of phytobiotic containing cinnamon oil on lipid status in the blood of the chickens ^1^.

		TC mmol/L	HDL mmol/L	LDL mmol/L	TAG mmol/L	NEFA µmol/L
G-C ^2^		3.97	1.47	2.344	0.78	34.25
CT-0.05		2.45 *	2.25 *	1.07 *	0.65 *	31.28
CT-0.1		2.43 *	2.44 *	0.882 *	0.54 *	27.47 *
CT-0.25		2.29 *	2.87 *	0.338 *	0.41 *	23.04 *
PT-0.05		2.74 *	2.47 *	1.122 *	0.74	32.47
PT-0.1		2.28 *	2.59 *	0.576 *	0.57 *	30.28
PT-0.25		2.07 *	2.75 *	0.236 *	0.42 *	24.28 *
SEM		0.082	0.027	0.08	0.017	27.47 *
Dosage effect (D)	0.05 (ml/L)	3.59	2.36	1.096	0.695	31.38 ^a^
	0.1 (ml/L)	3.36	2.52	0.729	0.555	28.88 ^a^
	0.25 (ml/L)	3.18	2.81	0.287	0.415	23.66 ^b^
Time effect (T)	CT	3.39	2.52	0.760	0.530	27.26
	PT	3.36	2.60	0.645	0.577	29.01
*p*-value						
G-C vs. all other		0.018	0.017	0.041	0.014	0.007
D effect		0.068	0.057	0.072	0.038	0.034
T effect		0.032	0.042	0.091	0.053	0.828
DxT interaction		0.335	0.374	0.587	0.502	0.474

* Means within the same column differ significantly from the control at *p* ≤ 0.05. ^a,b^ Means within the same column differ significantly (*p* ≤ 0.05) according to Newman–Keuls mean comparison (only in the case of significant DxT interaction). ^1^ Data represent mean values of seven replications per treatment. SEM = SD divided by the square root of the replication number, *n* = 7. ^2^ Designations for the treatments as for Table 2.

**Table 6 animals-11-00399-t006:** Effect of the level and duration of application of the phytobiotic containing cinnamon oil on enzymes activity in the blood of the chickens ^1^.

		LDH U/L	ALP U/L	GGT U/L	CK U/L	HBDH U/L	AC U/L
G-C ^2^		497.1	625.7	17.22	547.98	178.55	1.44
CT-0.05		387.3 *	678.6	14.89 *	479.36 *	147.47 *	1.58 *
CT-0.1		318.25 *	628.1 *	15.28	348.89 *	123.44 *	1.45
CT-0.25		231.7 *	558.2 *	14.58 *	389.89 *	127.47 *	1.47
PT-0.05		425.9 *	658.2 *	15.89	436.89 *	159.41	1.27 *
PT-0.1		414.08 *	698.3 *	16.17	447.89 *	160.47 *	1.25 *
PT-0.25		447.4 *	578.4 *	13.89 *	447.78 *	147.28 *	1.34
SEM		3.25	27.69	0.17	28.74	4.99	0.17
Dosage effect (D)	0.05 (ml/L)	406.6	668.4	15.39	458.13 ^a^	153.44	1.425
	0.1 (ml/L)	366.13	663.2	15.79	398.39 ^b^	141.96	1.35
	0.25 (ml/L)	339.55	568.3	14.24	418.84 ^a^	137.38	1.405
Time effect (T)	CT	312.42 ^b^	621.63	14.92	406.05 ^b^	132.79 ^b^	1.50
	PT	429.1 ^a^	644.97	15.32	473.56 ^a^	155.72 ^a^	1.29
*p*-value							
G-C vs. all other		<0.001	<0.001	0.034	0.021	0.028	0.025
D effect		0.003	0.059	0.321	<0.001	0.021	0.058
T effect		<0.001	<0.001	0.942	0.003	0.049	0.073
DxT interaction		0.006	0.081	0.810	0.042	0.071	0.087

* Means within the same column differ significantly from the control at *p* ≤ 0.05. ^a,b^ Means within the same column differ significantly (*p* ≤ 0.05) according to Newman–Keuls mean comparison (only in the case of significant DxT interaction). ^1^ Data represent mean values of seven replications per treatment. SEM = SD divided by the square root of the replication number, *n* = 7. ^2^ Designations for the treatments as for Table 2.

**Table 7 animals-11-00399-t007:** Effect of the level and duration of application of the phytobiotic preparation containing cinnamon oil on the immune status of the blood of the chickens ^1^.

		Lysozyme (mg/L)	Phagocytic Cells (%)	Phagocytic Index	Nitroblue Tetrazolium Test (%)	Interleukin 6 (pg/mL)	Immunoglobulin A (mg/mL)
G-C ^2^		1.15	34.24	4.14	20.87	0.120	0.697
CT-0.05		2.07 *	38.78	5.48	21.52	0.155 *	0.657
CT-0.1		2.18 *	42.47 *	5.56	25.78	0.141 *	0.765 *
CT-0.25		2.42 *	45.74 *	6.78 *	26.85	0.121	0.942 *
PT-0.05		2.08 *	41.07	5.66	21.25	0.142 *	0.682
PT-0.1		2.21 *	43.11 *	5.97	22.89	0.149 *	0.721 *
PT-0.25		2.29 *	42.04 *	6.35 *	24.66	0.142	0.987 *
SEM		0.03	0.27	0.07	0.14	0.08	0.13
Dosage effect (D)	0.05 (ml/L)	2.08	39.93	5.12	21.39	0.149	0.670 ^b^
	0.1 (ml/L)	2.20	42.79	5.77	24.34	0.145	0.743 ^b^
	0.25 (ml/L)	2.36	43.89	6.57	25.76	0.132	0.965 ^a^
Time effect (T)	CT	2.22	42.33	5.94	24.72	0.139	0.788
	PT	2.19	42.07	5.99	22.93	0.144	0.778
*p*-value							
G-C vs. all other		0.041	0.034	0.038	0.947	0.014	0.031
D effect		0.029	0.654	0.325	0.514	<0.001	0.031
T effect		0.187	0.231	0.308	0.217	0.051	0.054
DxT interaction		0.521	0.314	0.509	0.228	0.068	0.137

* Means within the same column differ significantly from the control at *p* ≤ 0.05. ^a,b^ Means within the same column differ significantly (*p* ≤ 0.05) according to Newman–Keuls mean comparison (only in the case of significant DxT interaction). ^1^ Data represent mean values of seven replications per treatment. SEM = SD divided by the square root of the replication number, *n* = 7. ^2^ Designations for the treatments as for Table 2.

**Table 8 animals-11-00399-t008:** Effect of the level and duration of application of the probiotic on the growth performance of the chickens ^1^.

		Body Weight (kg/Bird, 1–42 Day)				FCR (kg/kg, 1–42 Day)	Mortality Rate (Birds)
		1 Days	14 Days	35 Days	42 Days		
G-C ^2^		0.046	0.451	2.017	2.654	1.724	3
CT-0.05		0.045	0.452	2.017	2.666	1.717	2
CT-0.1		0.046	0.463	2.064	2.716 *	1.715 *	1
CT-0.25		0.045	0.457	2.148	2.735 *	1.712 *	1
PT-0.05		0.045	0.437	2.019	2.658	1.718	2
PT-0.1		0.044	0.441	2.112	2.681	1.717	2
PT-0.25		0.046	0.457	2.107	2.699 *	1.716 *	1
SEM		0.003	0.004	0.072	0.024	0.087	-
Dosage effect (D)	0.05 (ml/L)	0.045	0.445	2.018	2.662	1.718	-
	0.1 (ml/L)	0.045	0.452	2.088	2.699	1.716	-
	0.25 (ml/L)	0.046	0.457	2.128	2.717	1.714	-
Time effect (T)	CT	0.045	0.457	2.076	2.705	1.715	-
	PT	0.045	0.445	2.079	2.679	1.717	-
*p*-value							
G-C vs. all other		0,157	0.038	0.028	0.025	0.031	-
D effect		0.241	0.041	0.009	0.024	0.053	-
T effect		0.471	0.051	0.031	0.042	0.071	-
D × T interaction		0.587	0.142	0.141	0.428	0.063	-

* Means within the same column differ significantly from the control at *p* ≤ 0.05. ^1^ Data represent mean values of seven replications per treatment. SEM = SD divided by the square root of the replication number, *n* = 7. ^2^ Designations for the treatments as for Table 2.

**Table 9 animals-11-00399-t009:** Effect of the level and duration of application of the phytobiotic preparation containing cinnamon oil on the microbiological analysis of the jejunal contents and on measurements of jejunal villi and crypts in the broiler chickens ^1^.

		Total Number of Fungi (CFU/g)	Total Number of Aerobic Bacteria (CFU/g)	Total Number of Coliform Bacteria (CFU/g)	Length of Jejunal Villi (μm)	Depth of Jejunal Crypts (μm)
G-C ^2^		457	310,548	436,525	1014.25	208.75
CT-0.05		168 *	1,379,456 *	287,698 *	1174.10 *	247.45 *
CT-0.1		145 *	1,746,437 *	154,774 *	1258.84 *	249.78 *
CT-0.25		82 *	1,848,534 *	128,733 *	1318.98 *	274.74 *
PT-0.05		212 *	1,554,325 *	347,478 *	1012.74 *	217.48 *
PT-0.1		189 *	1,365,258 *	208,475 *	1169.74 *	246.31 *
PT-0.25		114 *	1,025,147 *	158,698 *	1198.14 *	253.02 *
SEM		15.25	58.36	25.36	58.25	16.17
Dosage effect (D)	0.05 (ml/L)	190	1,466,891 ^b^	317,588	1093.42	232.47
	0.1 (ml/L)	167	1,555,848 ^a^	181,625	12,414.29	248.05
	0.25 (ml/L)	98	1,436,841 ^b^	143,716	1258.56	263.88
Time effect (T)	CT	131.67	1,658,142 ^a^	190,402	1250.64	257.32
	PT	171.67	1,314,910 ^b^	238,217	1126.87	238.94
*p*-value						
G-C vs. all other		<0.001	<0.001	<0.001	0.002	0.003
D effect		0.002	<0.001	<0.001	0.027	0.024
T effect		<0.001	<0.001	<0.001	0.036	0.075
DxT interaction		0.07	0.008	<0.001	0.214	0.130

* Means within the same column differ significantly from the control at *p* ≤ 0.05. ^a,b^ Means within the same column differ significantly (*p ≤* 0.05) according to Newman–Keuls mean comparison (only in the case of significant DxT interaction). ^1^ Data represent mean values of seven replications per treatment. SEM = SD divided by the square root of the replication number, *n* = 7. ^2^ Designations for the treatments as for Table 2.

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
