# Peer review of "The Effect of Administration of a Phytobiotic Containing Cinnamon Oil and Citric Acid on the Metabolism, Immunity, and Growth Performance of Broiler Chickens"

_animals, 2021, doi:10.3390/ani11020399_

Round 1
Reviewer 1 Report
Dear Authors,
The subject matter of the paper “The effect of administration of a phytobiotic containing cinna-mon oil on the metabolism, immunity and growth performance of broiler chickens” is still relevant and important for the producers reducing the usage of antibiotics in poultry production. What deserves a particular recognition, is the comprehensive investigation of the activity of the additive used in the digestive tract of the birds. There are no methodological remarks I would like to make, however the authors are requested to update the literature presented:
- Smulikowska, S.; Rutkowski, A. (Eds). Recommended Allowances and Nutritive Value of Feedstuffs. Poultry Feeding Standards (in Polish). 4th Edition. The Kielanowski Institute of Animal Physiology and Nutrition, PAS, Jabłonna (Poland), 2005. – a new 2018 version is available
and change or delete the item:
Mazanowski, A. Modern production of broiler chickens. Gietrzwałd (Poland), 2011. – it is not an original work and it is not publicly available.
Author Response
COMMENTS TO THE REVIEWER 1
Thank you for the review; we have responded to the comments in the text. Fragments that have been modified or added are marked with a different color. Irrelevant references have been removed or replaced with appropriate ones. Missing references have been added.
Most of the revisions addressing the review have been made directly next to the Reviewer's comments in the text of the manuscript. They are marked in red.
Reviewer 1 wrote:
Dear Authors,
There are no methodological remarks I would like to make, however the authors are requested to update the literature presented:
- Smulikowska, S.; Rutkowski, A. (Eds). Recommended Allowances and Nutritive Value of Feedstuffs. Poultry Feeding Standards (in Polish). 4th Edition. The Kielanowski Institute of Animal Physiology and Nutrition, PAS, Jabłonna (Poland), 2005. – a new 2018 version is available
and change or delete the item:
Mazanowski, A. Modern production of broiler chickens. Gietrzwałd (Poland), 2011. – it is not an original work and it is not publicly available.
Answers:
Yes, we would like to thank the Reviewer for pointing out thatthere is indeed a newer version of the feeding norms for poultry. From now on, we will use the 2018 edition.
Regarding the position of literature: Mazanowski, A. Modern production of broiler chickens. Gietrzwałd (Poland), 2011, indeed the Reviewer is right that it is not an original work and is not publicly available. An editorial error has crept in. As suggested by the reviewer, we have replaced this one from the text.

Reviewer 2 Report
This manuscript determined the effect of administration of a phytobiotic containing cinnamon oil on the metabolism, immunity and growth performance of broiler chickens. This is a well-designed study. The experiment used 980 chicks for 7 treatments for a 42-days trial. The simple size is enough for the performance data. The presented the results is really bad. However, there are many issues need to be revised before the publication.
- The peer-reviewed manuscript do not have line in each papers. It is hard for the reviewers to do the comments.
- The Simple summary section could be generally described the findings. However, the Abstract section need to specified the details of the results. Such as, what kind of performance, intestinal, immunity parameters changes? And how much changes? P value etc.
- Page 2, “3000 mg/L cinnamon oil and 150,000 mg/L citric acid”, corrected to “3 mg/L cinnamon oil and 150 g/L citric acid”. Please check all the rest. Meanwhile, the dose of 150 g/L citric acid looks very high? Please check it.
- Table 4, in the group of CT-0.25, each birds consumption of 248.7 g citric acid, which means around 6 g/day. This looks very high. Please explain or citation reference for it.
- Table 4, are UA, BIL, CREAT parameters belong to antioxidant status?
- Table 6, It is strange to put enzymes activity are presented in a table. Table 3 also has AST and ALT activities. The authors need to classified the same function data to a table or group.
- Please re-organized the boold parameters according to their functions.
- Table 8, why not put the most important data in the first?
- Please explanation of the FCR with the intake of citric acid.
Author Response
COMMENTS TO THE REVIEWER 2
Thank you for the review; we have responded to the comments in the text. Fragments that have been modified or added are marked with a different color. Irrelevant references have been removed or replaced with appropriate ones. Missing references have been added.
Most of the revisions addressing the review have been made directly next to the Reviewer's comments in the text of the manuscript. They are marked in blue.
Reviewer 2 wrote:
This manuscript determined the effect of administration of a phytobiotic containing cinnamon oil on the metabolism, immunity and growth performance of broiler chickens. This is a well-designed study. The experiment used 980 chicks for 7 treatments for a 42-days trial. The simple size is enough for the performance data. The presented the results is really bad. However, there are many issues need to be revised before the publication.
The peer-reviewed manuscript do not have line in each papers. It is hard for the reviewers to do the comments.
Answer: We apologize for our oversight. The reviewer is right that the lack of line numbering makes it difficult to review the manuscript. We corrected the error and introduced continuous line numbering to the text.
Reviewer
- The Simple summary section could be generally described the findings. However, the Abstract section need to specified the details of the results. Such as, what kind of performance, intestinal, immunity parameters changes? And how much changes? Pvalue etc.
Answer: The Reviewer is right that the Abstract part should be supplemented with more specific information on the research results obtained. We supplemented the information in the Abstract section of our manuscript.
Reviewer
- Page 2, “3000 mg/L cinnamon oil and 150,000 mg/L citric acid”, corrected to “3 mg/L cinnamon oil and 150 g/L citric acid”. Please check all the rest. Meanwhile, the dose of 150 g/L citric acid looks very high? Please check it.
Answer: We have indeed entered the wrong order of magnitude for the amount of citric acid. Thank you for your attention. The preparation we used, containing: 3000 mg/L cinnamon oil and 150 mg/L citric acid, is a commercial preparation, and the information about the chemical content of the preparation was obtained from the manufacturer. The error has been corrected and the calculations have been corrected.
Reviewer
- Table 4, in the group of CT-0.25, each birds consumption of 7 g citric acid, which means around 6 g/day. This looks very high. Please explain or citation reference for it.
Answer: The reviewer is right, an typing error appeared in the table, the unit in which the consumption of citric acid was assessed is mg / bird. Thank you for paying attention. The error was corrected.
Reviewer
- Table 4, are UA, BIL, CREAT parameters belong to antioxidant status?
Answer: Uric acid, bilirubin and creatinine belong to the so-called hydrophilic low molecular weight antioxidants. They are part of the antioxidant system responsible for breaking the chain of free radical reactions at the propagation stage. This system also includes other antioxidants, such as vitamins A, C and E, glutathione, carnitine and flavonoids.
Reviewer
- Table 6, It is strange to put enzymes activity are presented in a table. Table 3 also has AST and ALT activities. The authors need to classified the same function data to a table or group. Please re-organized the boold parameters according to their functions.
Answer: The inclusion of AST and ALT in the group of biomarkers indicating the intensification of redox reactions in the body is not accidental. The enzymes AST and ALT belong to the so-called liver profile, and their activity proves the work of this organ. Most of the detoxification reactions take place in the liver, which also include reactions to inactivate free radicals to prevent oxidative stress. Oxidative stress caused by the presence of free radicals may cause damage to the work and structure of the liver cell. Based on the foregoing, we believe that this allocation of blood indicators to the tables is appropriate.
Reviewer
- Table 8, why not put the most important data in the first?
Answer: The authors considered all the results to be equally important, and the order in which they are presented results from certain dependencies. The indicators of growth performance of the chickens were only included in Table 8, i.e. after illustrating the biochemical, antioxidant and immunological indicators of blood, due to the fact that the authors assumed that the growth performance indicators are a consequence of metabolic and immune reactions, which taking place in the chickens body. Also in the title, growth performance indicators it were placed last, for the same reason.
Reviewer
- Please explanation of the FCR with the intake of citric acid.
Answer: The available literature contains information that citric acid increases feed intake, which has not been reported in our own research. Most likely, in our study, cinnamon oil had a much greater impact on feed intake and FCR than acetic acid. The beneficial effect of citric acid, however, manifests itself in improving the antioxidant status (citric acid forms complex compounds with metal cations of variable valence, such as iron and manganese, limiting their pro-oxidative effect), as noted in our research.

Reviewer 3 Report
This paper described the “The effect of administration of a phytobiotic containing cinnamon oil on the metabolism, immunity and growth performance of broiler chickens”. The paper is well written. However, I see some major issues that should be resolved before publishing this paper: Major: 1. The hypothesis is missing. The authors would like to investigate the optimal dosage and duration of phytobiotics. But why and how the dosage (0.05-0.25 g/L) and duration (1-7, 15-21, 29-35 d) tested in this study should be mentioned. 2. In the introduction section, the current findings about supplementation of cinnamon oil and citric acid, or both in broilers should be reviewed. 3. The effects of different dosages and duration of phytobiotics in broilers should be compared with other studies in the Discussion section. 4. In the Discussion section, the authors described that “A decrease in the number of coliforms and Bacillus subtilis in the intestinal contents following the use of cinnamon oil has been reported by Gupta et al. [27] and Abramowicz et al. [28].” Bacillus subtilis is widely accepted as probiotic and beneficial bacteria in the gut of poultry. Whether Bacillus subtilis in the jejunal contents is still affected by phytobiotics in this study? Minor: 1. The phytobiotics not only contain cinnamon oil, but also citric acid. The title should be modified. 2. In the Abstract: ….PT-0.05, PT-0.5, and PT-0.25 received the “probiotic” in the same amounts…. The typo should be corrected.Author Response
COMMENTS TO THE REVIEWER 3
Thank you for the review; we have responded to the comments in the text. Fragments that have been modified or added are marked with a different color. Irrelevant references have been removed or replaced with appropriate ones. Missing references have been added.
Most of the revisions addressing the review have been made directly next to the Reviewer's comments in the text of the manuscript. They are marked in green.
Reviewer 3 wrote:
This paper described the “The effect of administration of a phytobiotic containing cinnamon oil on the metabolism, immunity and growth performance of broiler chickens”. The paper is well written. However, I see some major issues that should be resolved before publishing this paper:
Major:
Reviewer
- The hypothesis is missing. The authors would like to investigate the optimal dosage and duration of phytobiotics. But why and how the dosage (0.05-0.25 g/L) and duration (1-7, 15-21, 29-35 d) tested in this study should be mentioned.
Answer: The reviewer is right, there is indeed no word in the hypothesis about the dosage of the preparation and the time of its application. The phytobiotic preparation tested in the presented experiment is a commercial product. The manufacturer suggests using it as a nutritional supplement intended for poultry and assumes a dosage of the preparation in the amount of 0.1-0.2 mL/L of water, with the possibility of using it throughout the rearing period. Due to the fact, that in large herd farms, for which the preparation is also recommended, the costs are carefully monitored, it was decided to check what health and production effect would be achieved by using the dose recommended by the manufacturer (0.1 mL/L) compared to the lower dose 0 , 05 mL/L and more than 0.2 mL/L, than recommended by the manufacturer. The same goes for the application time. It was also taken into account that a higher dose of the product used as an addition to the diet does not always mean obtaining better health effects or more satisfying rearing results. The experiment used continuous (for 42 days of rearing) and periodic (selected rearing periods) application. In the second case, an attempt was made to assess the effect of the preparation, possibly reducing the costs of its use. In the case of periodic application, a scheme was used consisting of alternating application of phytobiotic (one week of application, one week of break). In practice, in the first week of rearing (1-7 day), the preparation was administered to the water, followed by a week off; then, again the phitobiotic was applicated (15-21 day) and followed by a week off. In the time 29-35 day, again was an application of phytobiotic. The periodic administration of the preparation was adapted to the feeding periods of the birds (start of the starter phase: 1-7 days, end of the starter phase: 15-21 days and end of the grower phase: 29-35 days.
An excerpt has been added to the Introduction part of the manuscript:
The phytobiotic preparation used in the experiment is a commercial preparation, for poultry. The manufacturer assumes a dosage of the preparation in the amount of 0.1-0.2 mL/L of water, with the possibility of using it throughout the rearing period. For the purposes of the experiment, it was decided to check what health and production effect would be achieved by using the dose recommended by the manufacturer (0.1 mL/L), compared to a dose lower than 0.05 mL/L and greater than 0.2 mL /L, than recommended by the manufacturer. Moreover, the introduction of two modes of administration of the test product to the experimental scheme: continuous and periodic, was aimed at checking whether the periodic application of phytobiotic supplement would bring comparable effects as the continuous application. The periodic application of the preparation was adapted to the feeding periods of the birds (start of the starter phase: 1-7 days, end of the starter phase: 15-21 days and end of the grower phase: 29-35 days.
Reviewer
- In the introduction section, the current findings about supplementation of cinnamon oil and citric acid, or both in broilers should be reviewed.
An excerpt has been added to the Introduction part of the manuscript:
In addition to phytogenic additives, poultry rearing can also use organic acids, alone or combined with phytobiotics. According Fascina et al. (2012) the addition of cinnamon oil and citric acid improve the nutrient digestibility of the diet and replace the growth- promoting antibiotics. The results of the research by Pirgozliev et al., 2008; Ao et al., 2009 showed that combination of cinnamon oil and citric acid has a positive effect on the production performance of poultry as it lowers the pH of the intestinal contents and increases intolerance of bacterial growth to pH changes (Pirgozliev et al., 2008; Ao et al., 2009). The result is better gut health, better intestinal villi integrity, and maximum nutrient absorption (Dibner & Buttin, 2002). Additionally, undissociated organic acids can penetrate the lipid membrane of the bacterial cell and lower the pH inside it, which leads to the death of the bacterial cell (Ricke, 2003). Research by Ao et al. (2009) and Rizzo et al. (2010) showed that citric acid alone can also increase the ability to metabolize feed coefficients of nutrient metabolizable, which results in improved intestinal villi integrity and increased lipid absorption from the diet. According to Fascin et al. (2012) phytobiotics significantly better improve the health of chickens in the initial phase of feeding starter, thanks to the higher production of pancreatic enzymes. Acidifying, on the other hand, improves nutrient metabolism in the growth phase much better, according to Fascin et al. (2012) due to the fact by more intestinal contents. According Fascin et al. (2012) opinion, combination of cinnamon oil and citric acid in the diet supports muscle growth, without much wear and tear of renewing tissues.
Reviewer
- The effects of different dosages and duration of phytobiotics in broilers should be compared with other studies in the Discussion section.
The inclusion of cinnamon oil in the diet in amounts 250 or 500 mg/kg of feed improve to activity of antioxidant mechanisms in Japanese quails (Simsek et al. 2013). Chowdhury et al. (2018) reported that dietary supplementation with 300 mg cinnamon bark oil on kg of feed increased villi height in the duodenum or jejunum.
Answer: We agree with the Reviewer's suggestion that the introduction of information on the amount of preparation or compound used will give a better picture of the comparison of the effectiveness of their impact on the organism of broiler chickens. Relevant information is provided in the text.
Reviewer
- In the Discussion section, the authors described that “A decrease in the number of coliforms and Bacillus subtilis in the intestinal contents following the use of cinnamon oil has been reported by Gupta et al. [27] and Abramowicz et al. [28].” Bacillus subtilis is widely accepted as probiotic and beneficial bacteria in the gut of poultry. Whether Bacillus subtilis in the jejunal contents is still affected by phytobiotics in this study?
Answer: Indeed, the reviewer is right. There was an error in the work resulting from the appearance of the so-called mental shortcuts. Research by Abramowicz et al. (2020) concerned the comparison of the effectiveness of the effect of a preparation containing cinnamon oil (0.25 ml / L), in the context of the effect of an additive containing Bacillus subtilis on intestinal morphometry, while and in the studies by Gupta et al. cinnamon oil limited the proliferation of Bacillus subtilis in the MIC test, not in the jejunal. Information regarding this threadhas been removed from the text. Thank you for this attention.
Reviewer
Minor:
- The phytobiotics not only contain cinnamon oil, but also citric acid. The title should be modified.
Answer: The title was changed as suggested by the Reviewer.
Reviewer
- In the Abstract: ….PT-0.05, PT-0.5, and PT-0.25 received the “probiotic” in the same amounts…. The typo should be corrected.
Answer: Thank you very much for paying attention. This is a typo error. The name was corrected in the text.

Round 2
Reviewer 2 Report
No further comments.